# Internet Development, Level of Industrial Synergy, and Urban Innovation

**Hongxia Zhang, Zixuan Sun \*, Ehsan Elahi and Yuge Zhang**

School of Economics, Shandong University of Technology (SDUT), Zibo 255022, China;
zghx003@sdut.edu.cn (H.Z.); ehsanelahi@nuist.edu.cn (E.E.); zygmax001@gmail.com (Y.Z.)
**\*** Correspondence: szxmax001@gmail.com

**Abstract:** Innovation increases total factor productivity and leads to economic development. Based on panel data of 284 prefecture-level cities from 2001 to 2018, the current study uses a dynamic panel data model to empirically test the global and heterogeneous effects of internet development and industrial synergy on the level of urban innovation. Results found that the internet development significantly promoted the urban innovation level, and industrial collaboration was found to have a positive impact on the urban innovation level. Moreover, it was determined that the regulatory effect of the internet promoted industrial collaboration to improve the level of urban innovation. Variations in the impact of internet development and the industrial collaboration level on the urban innovation level were found in cities. Particularly, the impact of internet development and the industrial collaboration level on the urban innovation level in high-level cities was less significant. A positive role of the government is required to improve the level of urban innovation. Particularly, it is required to connect enterprises with universities to exchange scientific and technological knowledge, thereby improving urban innovation.

**Keywords:** internet; urban innovation; industrial synergy; impact analysis; China

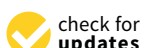



## 1. Introduction

### 1.1. Background

The Chinese high-speed growth was directly dependent on technological progress [1]. In recent years, the Chinese government suggested new normal development concepts and innovative driving development strategies to promote economic growth. The rapid development of global internet technology was organically accompanied by the current Chinese economic reform using internet technology to promote the efficiency and output quality of industrial organizations. It will favorably support Chinese innovation development. Urban innovation is an important reflection of Chinese economic growth. The degree of deepening and integration of urban industries will not only improve urban production efficiency, but also be an important driving force for urban innovation. For example, the sharing mechanism and technology spillover effect formed by the division and cooperation of industries in cities can promote the exchange of knowledge and technology among industries and promote the technological upgrading among industries, so as to realize the innovation of the urban scientific and technological level. Therefore, it is important to determine the efficiency of industrial organizations through the development of industry synergy. Afterward, it is necessary to determine whether or not the development of the internet affects the innovation of industrial collaboration.

The productivity of the industry depends on labor and capital change [2]. The industrial integration infiltration can be achieved by promoting industrial upgrading and forming collaborative innovation practices. As a carrier of innovation and knowledge flow, the internet has developed to reduce information transaction costs, improve transaction efficiency, and boost enterprises to actively conduct innovative research and development activities. The development of the internet promotes production factors in high-speed flow

between industries and realizes the improved efficiency of the industrial division of labor. Further, it can promote the development of the industry to promote the level of labor, and urban innovation [3].

### 1.2. Development and Innovation of the Internet

The internet integrated index has been used in previous studies as an indicator through a linear regression model, space Dubin model, panel data analysis, and threshold regression. These models empirically tested whether the internet can improve the technology innovation level [4]. Additionally, there is a nonlinear impact of the internet on the innovation level [5]. Previous studies also found a relationship between the internet and innovation from theoretical perspectives [6]. The study reported a significant forward impact on the innovation of the region. Neighborhood innovation also has a significant driving role, and the internet can promote enterprise innovation activities through technology exchange, advertising, and marketing [7]. The internal information transferred mechanism optimization and internet knowledge overflow effects can promote inclusive innovation and innovative performance enhancements [8], and the internet can also affect innovative efficiency through the association and interaction between the industries. However, Jiao et al. [9] found heterogeneity and time lag. It was reported in many studies that internet development has a positive role in innovation [10,11].

### 1.3. Industrial Collaboration and Innovation

Previous literature analyzed the impact of industrial collaboration on enterprise innovation. Yang et al. [12] believed that the corporate innovation ability is consistent with the manufacturing and service industry. Taddeo et al. [13] determined that the industry improved the innovation capability of enterprises through transaction costs and research and development incentives. Some studies focused on external perspectives, and particularly estimated the impact of industry collaboration on regional innovation [14,15], and other studies discovered that its impact on innovation has regional differences [16,17]. Most of the previous studies mainly discussed the linear and nonlinear relationships of internet development and regional innovation efficiency based on a microscopic perspective, and it was found that the internet was less impacted on urban innovation levels from the perspective of industrial collaboration. To the best of our knowledge, limited studies have considered the large heterogeneity of urban grade and urban size. Based on the establishment of industrial collaboration and the development of urban innovation, the current study focuses on the heterogeneity characteristics of urban innovation, and it is based on urban grading. Furthermore, it emphasizes prefecture city-level data and prefecture-level market panel data to explore the development of the internet. The determination of the relationship between industrial collaboration and the urban innovation level provides a broader idea to enhance the level of urban innovation.

## 2. Theoretical Framework and Construction of the Research Hypothesis
### 2.1. Innovation Effect of Internet Development

Internet development can effectively reduce transaction costs and unnecessary intermediate links [18]. It improves the. transaction efficiency, and promotes efficient integration configuration of resources, and may have an impact on innovation [19]. On the one hand, the development of the internet can save innovation process information acquisition costs and transaction costs [20]. Innovation efficiency is affected by the information transfer method, and internet development effectively reduces the time lag and asymmetry of information transmission between innovative bodies, which is convenient for innovative entities to obtain information and knowledge [21]. The internet has the characteristics of openness and sharing, by playing economic effects, such as an agglomeration economy and network economy [22]. It can effectively improve the transaction efficiency and reduce the application of assets of research and development activities and research and development costs [23]. Furthermore, it enhances the confidence and enthusiasm of innovative entity

research and development and promotes innovation activities. On the other hand, the internet platform can enhance the matching efficiency of innovative resources and promote the efficient integration of resource endowment. The internet enhances the internal resources data matching to achieve resource allocation efficiency based on their own platform [24]. Therefore, we proposed the following hypothesis:

**Hypothesis 1 (H1).** *Internet development can enhance urban innovation.*

*2.2. Industrial Collaborative Division Innovation Effect*

Collaboration in the manufacturing and productive service industry enhanced the communication efficiency of knowledge and technology [25]. It promoted the high-speed flow of people and other resource elements in the industrial flow and prompted frontier technology and philosophy in regional communication. According to the professional externality theory, the cause of regional specialization is that the economic policies and feature endowments in the region provide favorable conditions for such companies. The regional professional synergistic gathering makes the company give full play to knowledge spillage effects, thereby forming the source of regional innovation [26]. Regional specialization and self-cooperation are a smooth channel for innovation activities, management philosophy, and trade-edge field information, which is convenient for the exchange of knowledge and technology between enterprises. The knowledge overflows affect the specialized division of labor, which provides strong conditions for industrial advantage competition [27]. Once the advantageous industry is formed, it will attract the government to increase its support for its innovation activities, which results in greater demand in the process of improving innovation capabilities [28]. Furthermore, the development of the upper and lower related industries and the formation of the industrial chain system means the multi-industry collaboration promotes the level of innovation [29]. Combined with consumer utility theory, the production needs of diversified products can meet the utility of consumers to maximize the utility, as well as changes in production costs and new technological innovations. This will try to seek different industries and diversified services in the downstream related to the industry. Based on the industrial diversification co-collector platform, and industrial knowledge, the technology overflow is more convenient, and urban innovation levels also increase. Therefore, we propose the following hypothesis:

**Hypothesis 2 (H2).** *Industrial specialization and diversified self-division of labor improve the level of urban innovation.*

*2.3. Internet Development, Industrial Collaboration Level, and Urban Innovation in the Inner Connection*

Internet development can construct a certain advantage in terms of the information acquisition cost and transaction efficiency, increasing the enthusiasm of enterprise research and development and innovation activities [30]. It affects the level of innovation in the city. The synergy between the productive service industry and the manufacturing industry can promote the flow of production factors, new technologies in industries, and inter-regional communication, which will promote urban innovation [31]. The high-speed mobile promotion industry of industrial element information improved the professional division of labor and production efficiency of the industry [32]. The enterprises can become time effective during innovation activities. Information enhances the innovation and initiative of the company, which results in innovation of the production technology to improve the production efficiency [33]. Based on the theoretical background, we constructed Hypotheses 3 and 4:

**Hypothesis 3 (H3).** *The development of the internet can have an adjustment effect to promote industrial collaboration.*

**Hypothesis 4 (H4).** *The development of the internet in different levels and industrial types results in differences in urban innovation.*

## 3. Materials and Methods

### 3.1. Source of Data and Sampling Procedure

The data were collected from China City Statistical Yearbook, China Statistical Yearbook, China Industrial Statistics Yearbook, China Tertiary Industry Statistical Yearbook, National Databases, and Municipal Statistical Yearbooks. To maintain the consistency of the statistical caliber in the selection of the time period, the data were collected from 2002 to 2019 from the yearbooks. The observation data were collected from 2001 to 2018. Due to the completeness and availability of data, as well as the changes in the administrative divisions and names of cities over the years, 284 Chinese cities were selected at the prefecture level and above as the sample observations, and the missing values of a small number of cities were filled by the interpolation method [34].

### 3.2. Measurement of Statistical Model

The given statistical model was used to approach the objectives of the study:

$$Inno_{it} = \alpha_0 + \alpha_1 Col_{i,t-1} + \alpha_2 Control_{i,t-1} + \alpha_4 Col_{i,t-1} \times \ln t_{i,t-1} + \sigma_t + \lambda_i + \varepsilon_{it} \quad (1)$$

where "*I*" and "*t*" indicate urban and year, respectively. Moreover, $Inno_{it}$ represents the city innovation level, $Col_{i,t-1}$ represents the various local grades of manufacturing and productive service industry coexistence, $\ln t_{i,t-1}$ represents the development level of the internet, and $Control_{i,t-1}$ represents the control variables at the prefecture-level city specifically including human capital (HUM), per capita production (PGDP), government science and technology expenditure (GOV), foreign-dependent degree (FDI), and traffic infrastructure (TRF). $\delta_t$ and $\lambda_i$ represent the year and urban individual fixation effects. $\varepsilon_{it}$ represents a random interference item that is assumed to be normally distributed with zero mean values and constant variance [35–37].

### 3.3. Selection and Measurement of Variables

#### 3.3.1. Evaluation of the Index System

The level of urban innovation (Inno) of local cities was measured using the urban innovation index. Following Pakes and Schankerman [38], the index was used for the patent renewal model as a reference to calculate the average value of patents of different ages as the value weighting coefficient of corresponding patents. Afterward, the patent value stock was obtained by summing up the patent value of different cities, and the total national patent value in 2001 was standardized to 100. There are few indexes of the internet development level (int). Most of the internet has been used to characterize the number of internet pages. The number of urban internet broadband access points has been used as a variable that measures the internet level [39]. According to the GM (1, N), the industrial synergy measurement model constructed by grey theory, the advantage of this coefficient is that it can effectively avoid the incompleteness and grey of sample information caused by randomness and uncertainty in the interaction between the manufacturing industry and service industry [40]. Referring to the classification standard for producer services formulated by the National Bureau of Statistics in 2015 and the national standard for industry classification, producer services are defined as industries with two-digit codes ranging from 51 to 63 and 68 to 78, including warehousing and postal industry, transportation industry, information transmission, computer services and software industry, wholesale and retail industry, finance industry, leasing and commercial services, scientific research, and technical services.

In the process of constructing the evaluation index system of the development level of the manufacturing industry and producer services, this study combined the connotation and attributes of the manufacturing industry and producer services and refers to the interactive evaluation index system of the two industries. The comprehensive development

evaluation index system of the two industries constructs the development level index system of the manufacturing industry and service industry from four aspects: 1 economic benefits; 2 industrial scale; 3 growth potential; and 4 social contribution. Specific indicators are given in Table 1.

**Table 1.** Evaluation index system of the comprehensive development level of the manufacturing industry and producer service industry.

| Primary Index | Secondary Index | Index Interpretation | Unit |
|---|---|---|---|
| Economic Performance | Labor productivity | GDP/number of employees | Chinese Yuan per person |
| | Investment in fixed assets | Total investment in fixed assets | 100 Million Chinese Yuan |
| Industrial Scale | GDP | Total GDP | 100 Million Chinese Yuan |
| | Number of business units | Total number of enterprises | Per |
| Growth Potential | Proportion of investment in total social investment | (Fixed asset investment/National fixed asset investment) × 100% | Percentage |
| | GDP growth rate | (GDP of the current year/GDP of the previous year − 1) × 100% | Percentage |
| Social contribution | Number of employees | Total number of employed persons | Ten thousand people |

To evaluate the index system of the comprehensive development level of the manufacturing industry and producer service industry, this paper weighted the index by the entropy method, and determined the coordination and coupling degrees. The estimation process can be given as:

The coupling degree "$C$" and coordination degree "$D$" of the two subsystems can be estimated using Equations (2) and (3), respectively:

$$C = 2[\frac{U \times G}{(U + G)^2}]^{\frac{1}{2}} \tag{2}$$

$$D = \sqrt{C \times T} \tag{3}$$

In Equation (2), the coupling degree "$C$" represents the degree to which the two subsystems affect each other through their respective coupling factors. When $C$ = V, and $0.3 < C \leq 0.7$, it indicates that the two subsystems are in a moderately coupled state. When $0.7 < C < 1$, it indicates that the two subsystems are in a highly coupled state. Similarly, when $C = 1$, it indicates that the two subsystems are fully coupled. In Equation (3), "$T$" is the comprehensive evaluation index that represents the overall synergy of the two subsystems, $T = \theta \times U + \gamma \times G$, $\theta + \gamma = 1$, and $\theta$ and $\gamma$ are the weights of the two subsystems. Referring to the existing literature, the weight of subsystem 1 is $\theta$ (=0.5), and the weight of subsystem 2 is $\gamma$ (=0.5).

### 3.3.2. Coordination Degree

The coordination degree "$D$" refers to the degree of mutual coupling between two subsystems. The coordination degree of the two subsystems is between 0 and 1 (Table 2). According to the size of coupling scheduling, the two industries can be divided into 10 levels, as shown in Table 2. According to the evaluation model, the entropy method was first used to calculate the weight of each indicator in the indicator system. Secondly, according to the estimation method of the coupling degree and coordination degree, we estimated the score "$U$", score "$G$", and coupling degree "$C$" of the two subsystems each year. Finally, according to the classification standard of the coupling coordination degree, the coupling coordination level of the two subsystems was determined.

**Table 2.** Coupling and coordination degree grades and classification level of the two major industries.

| Coordination Levels | Horizontal Classification | Coordination Degree |
|---|---|---|
| Extreme imbalance | | (0, 0.1) |
| Severe imbalance | Bud stage | (0.1001, 0.2) |
| Moderate disorder | | (0.2001, 0.3) |
| Mild disorder | | (0.3001, 0.4) |
| On the verge of maladjustment | Initial stage | (0.4001, 0.5) |
| Barely coordinated | | (0.5001, 0.6) |
| Primary coordination | Stable phase | (0.6001, 0.7) |
| Intermediate coordination | | (0.7001, 0.8) |
| Well-coordinated | Mature stage | (0.8001, 0.9) |
| Quality coordination | | (0.9001, 1) |

### 3.3.3. Control Variables

Based on the existing literature on the factors affecting the level of urban innovation, the article considers the following control variables: First, human capital (hum), which is characterized by the ratio of the number of higher education students to the urban population; second, the per capita GDP (pgdp), expressed as the ratio of GDP to the urban population; third, government expenditure on science and technology (gov) is characterized by the proportion of fiscal expenditure on science and technology in the total fiscal expenditure of each city; fourth, the degree of Foreign Direct Investment (fdi). Due to the incomplete data on the municipal districts, it is measured by the proportion of the actual use of foreign capital in the city at the level of the city in the year to the city's GDP; and fifth, the availability of traffic infrastructure (trf), as the index of road kilometers per capita in prefecture-level cities is low. We used the per capita urban road area of each city as a proxy variable.

### 4. Results and Discussion

#### 4.1. Results of Benchmark Regression

This paper used the ordinary least square (OLS) method, fixed effect (FE) model, and random effect (RE) model to perform regression analysis [41–45]. The regression results of the models in columns 2, 3, and 4 of Table 3 showed that the internet has a positive impact on the urban innovation level, which is consistent with the expected sign. The OLS method found the greatest promotion effect on the improvement of the city's innovation level. This means by increasing one unit of internet access, the innovation level of the city increased by 1.94%. The promotion effect of the internet is the smallest. With an increase in an additional unit of internet access, the innovation was increased by 1.13%. The regression results suggest that the innovation effect of internet development has a positive effect of promoting urban innovation (the finding is consistent with Hypothesis 1). Moreover, the coordination of manufacturing and producer services under the three regression results has a positive impact on the city's innovation level. These results are in line with the study of [39]. However, in addition to the results of OLS, the results of the other two regressions were not significant at 10%, and the coefficients were smaller than the influence coefficient of the internet. This indicates that the synergy between the manufacturing industry and the producer service industry can promote urban innovation to a certain extent. These results verified Hypothesis 2; however, it is generally weaker than the internet in terms of increasing intensity.

The control variables, such as human capital, government financial technology, expenditure, economic development level, and transportation infrastructure, have a significant positive impact on urban innovation. Nie et al. [46] also found similar findings. After introducing FDI, the regression coefficient changed with a negative sign. It indicates that FDI has a negative impact on the level of urban innovation, and it is not conducive to the improvement of the level of urban innovation. These results are consistent with Xiang and Chuanhai [47] and Li, S. et al. [48]. By combining the above-mentioned weak promotion

effect of the collaboration between manufacturing and productive services on the level of urban innovation, we may explain the following points. Firstly, the diversified externalities of the collaboration between the productive service industry and the manufacturing industry formed a variety of different fields and industries. After the introduction of FDI, the cost of obtaining information and transaction costs further increased. It reduced the initiative and enthusiasm of some enterprises in innovation, which is not conducive to the level of urban innovation. Secondly, the upstream and downstream collaborative-related industries within the cluster are at different stages of development, and the development concepts and production methods of different industries are heterogeneous. The absorption capacity of new technologies and new concepts was. brought about by the introduction of FDI. It caused a fault in the collaboration between related industries and then impacted the original state of synergy.

Furthermore, it is found that the cross-term regression results of industrial synergy and the level of internet development are significantly positive. It indicates that the development of the internet will effectively improve urban innovation in the collaborative environment of manufacturing and producer services. The development of the internet affects industrial synergy [49]. The influence of innovation showed a moderating effect. These results satisfied Hypothesis 3. As the carrier of information flow, the internet accelerates the efficient dissemination of information in the innovation system, breaks the space and time constraints of information circulation, and enables the collaborative division of labor to obtain time-sensitive and cutting-edge technologies and ideas at a lower cost. Original scattered innovation resources can be re-integrated. Resource allocation is further optimized, and the level of collaboration and division of labor between industries is also improved, thereby realizing urban innovation [50].

**Table 3.** Results of the benchmark regression.

| Variables | FE | OLS | RE |
|---|---|---|---|
| Col | 0.69 (0.47) | 0.78 * (0.71) | 0.42 (2.28) |
| Internet development | 1.54 *** (2.51) | 1.94 *** (3.38) | 1.13 *** (1.86) |
| Government science and technology expenditure | 0.55 *** (0.53) | 0.68 *** (0.56) | 0.42 *** (0.78) |
| Human capital | 0.39 *** (3.41) | 0.27 *** (5.61) | 0.47 *** (5.39) |
| Traffic infrastructure | 0.60 * (0.74) | 0.37 ** (0.52) | 0.54 * (0.53) |
| Per capital production | 0.21 *** (0.13) | 0.27 *** (0.14) | 0.20 *** (0.92) |
| Foreign direct investment | −1.62 (−0.78) | −1.38 (−0.80) | −2.15 (−0.84) |
| Col * Internet | 1.23 *** (1.39) | 1.50 ** (2.14) | 0.85 ** (1.93) |
| Constant | 0.57 *** (0.41) | 0.61 ** (0.60) | 0.55 *** (0.53) |
| N | 5112 | 5112 | 5112 |
| $R^2$ | 0.86 | 0.79 | 0.84 |

*, **, *** represents level of significance at 10%, 5%, and 1%, respectively. *t*-values are given in parentheses.

### 4.2. Analysis of Heterogeneity

Considering that there are cities of different levels and levels of development, the innovation effects of the internet development and industrial synergy levels may differ. We divided 284 city samples into first-tier cities, new first-tier cities, second-tier cities, third-tier cities, fourth-tier cities, and second-tier cities based on the "2020 City Business Charm Ranking List" published by China Business News. Five groups of samples from fifth-tier cities were analyzed. The results are shown in Table 4. According to the results of columns 2, 3, 4, 5, and 6, the internet and industrial synergy of the five groups of city samples have a significant positive impact on the level of urban innovation. The significance of new first-tier cities and second-tier cities is lower than that of third-tier cities and below. The possible explanation is that the first-tier cities, new first-tier cities, and second-tier cities have abundant element resources, resource allocation efficiency is high, and inter-industry collaboration capabilities are strong. Third-tier cities and below have a lack of economic material resources and low resource allocation efficiency, internet development, and industry. The level of collaboration

is the driving force for innovation. Through regression coefficients, it is found that the unit innovation that affects the synergy between the internet and the industry in first-tier cities, new first-tier cities, and second-tier cities is much greater than that of third-tier, fourth-tier, and fifth-tier cities. Combined with the analysis of other variables, the possible explanation is that the economic development level of first-tier cities, new first-tier cities, and second-tier cities is high, the transportation infrastructure is perfect, and the influx of a large number of foreign labor forces expands the local human capital pool and improves the efficiency of urban innovation [51]. At the same time, strong policy support and a high level of information technology promote the cooperation of advantages between industries, stimulate the initiative and enthusiasm of enterprise innovation, and thus produce a higher innovation effect [52]. This finding satisfies Hypothesis 4.

**Table 4.** Results of heterogeneity analysis.

| Variables | First-Tier Cities | New First-Tier Cities | Second-Tier Cities | Third-Tier Cities | Fourth and Fifth-Tier Cities |
|---|---|---|---|---|---|
| Col | 235.40 ** (0.24) | 195.85 * (1.97) | 63.22 ** (1.99) | 28.78 *** (0.63) | 3.37 *** (0.73) |
| Internet development | 2.31 * (0.57) | 1.25 * (0.83) | 1.06 * (0.95) | 0.19 ** (0.59) | 0.20 *** (0.26) |
| Government science and technology expenditure | 0.63 *** (0.53) | 0.42 *** (0.49) | 0.17 * (0.10) | 0.06 (0.03) | 0.01 * (0.00) |
| Human capital | 0.85 *** (0.47) | 0.13 * (0.07) | 0.05 * (0.06) | −0.11 * (−0.18) | −0.15 *** (−0.83) |
| Traffic infrastructure | 0.26 (0.06) | 0.25 (0.11) | 0.50 *** (0.30) | 0.00 (0.01) | 0.01 *** (0.03) |
| Per capital production | 0.31 *** (0.27) | 0.43 ** (0.71) | 0.48 *** (0.95) | 0.50 *** (0.18) | 0.22 *** (0.19) |
| Foreign direct investment | −0.41 *** (−0.53) | −0.53 * (−0.83) | −0.60 *** (−0.28) | −0.32 * (−0.79) | −0.56 *** (0.51) |
| Col * Internet | 162.55 *** (1.33) | 24.43 *** (0.99) | 18.63 *** (0.47) | 3.96 *** (1.11) | 7.27 *** (0.66) |
| Constant | −2.73 ** (−0.24) | 1.66 * (0.18) | 0.51 * (1.69) | 2.43 *** (0.56) | 2.566 *** |
| N | 72 | 270 | 540 | 1260 | 2970 |
| R² | 0.74 | 0.74 | 0.68 | 0.75 | 0.77 |

*, **, *** represents level of significance at 10%, 5%, and 1%, respectively. *t*-values are given in parentheses.

### 4.3. Endogenous Issue

We used the Hausman test for each variable and found that Prob>chi² = 0.00016, and the *p*-value is less than 5%. Therefore, the original hypothesis is rejected. The variables of the model identified the problem of endogeneity. To deal with endogeneity problems, the explanatory variables were treated with a lag period, and human capital, government fiscal expenditure, and other variables were added to control the possible missing variables. At the same time, the optimal fixed-effect model was selected for regression and estimation through Hausmann's test [53]. In most studies, the endogenous problem was treated with fixed-effects 2SLS model regression [54,55]. Considering the heteroscedasticity and autocorrelation of panel data, the GMM method was more appropriate and accurate than the 2SLS estimation result [56]. Furthermore, this paper chose GMM to estimate the lagging one-period model. The results are given in Table 5. After using GMM, the results of each variable and the main estimation did not change.

**Table 5.** Endogenous analysis.

| Variables | GMM | (1) | (2) |
|---|---|---|---|
| L | 1.15 *** (2.01) | | |
| Col | 0.63 (0.16) | 0.76 (0.52) | 0.69 * (0.81) |
| Internet development | 1.10 *** (0.51) | 1.23 ** (1.85) | 1.03 *** (0.94) |
| Government science and technology expenditure | 0.01 *** (0.16) | 0.62 *** (0.60) | 0.55 *** (0.72) |
| Human capital | 0.45 *** (0.36) | 0.44 *** (3.82) | 0.42 *** (3.36) |
| Traffic infrastructure | 0.01 * (0.19) | 0.68 *** (0.83) | 0.60 *** (0.72) |
| Per capital production | 0.03 *** (1.27) | 0.24 ** (0.14) | 0.49 ** (0.13) |
| Foreign direct investment | −0.63 (−0.16) | −1.82 * (−0.88) | −1.60 (−0.93) |
| Col * Internet | 0.83 *** (1.20) | 1.38 ** (1.56) | 1.21 ** (1.02) |
| Constant | −0.74 *** (−0.20) | 0.65 ** (0.46) | 0.57 *** (0.41) |
| N | 4828 | 5112 | 5112 |

*, **, *** represents level of significance at 10%, 5%, and 1%, respectively. *t*-values are given in parentheses.

*4.4. Determination of Robustness*

In the regression analysis, the city innovation index was used as an explanatory variable to indicate the level of urban innovation. The number of internet broadband access points was used as an explanatory variable to characterize the development of the internet. To further enhance the reliability, a substitution variable method was used to continue the verification. The specific method replaces the city's innovation index with the number of urban patent grants and replaces the number of internet broadband access points with the internet penetration rate and re-estimates the model. The results are given in Table 5. It is found that except for fdi, the estimated coefficients of each variable were significantly positive, and the unit innovation effect of internet development was weakened.

**5. Conclusions and Policy Implications**

Based on the panel data of 284 prefecture-level cities from 2001 to 2018, this paper used a dynamic panel data model to empirically test the global and heterogeneous effects of internet development and industrial synergy on the level of urban innovation. The improvement of the urban innovation level has an obvious promotion effect. An additional unit of internet access maximized the innovation level by 1.94%. At the same time, the impact of industrial synergy on the urban innovation level was positive, but the impact on the innovation level was weaker than that of the internet. The introduction of FDI inhibited the internet's promotion of the city's innovation level to a certain extent. The impact of urban internet development and industrial synergy at different levels on the city's innovation level was different. The effect of the internet and industrial synergy in cities and second-tier cities on the improvement of urban innovation was weaker than that of third-tier and lower-tier cities, but the innovation and improvement effect of each unit brought by the internet and industrial synergy in these three groups of cities was better than that of third-tier and fourth- and fifth-tier cities.

Based on the study findings, we propose the following policy implications: Under the guidance of improving the city's innovation ability, differentiated strategies should be adopted for cities of different levels and development levels. It is necessary to implement precise industrial policies for prefecture-level cities by local conditions, and characteristics of the manufacturing industries of different cities to support the development of related producer services in a targeted manner, and to promote positive interaction and collaboration. For example, for cities around the Yellow River and other waters, the government should encourage complementary advantages with surrounding cities, learn from the urban development model of the Yangtze River Delta and the Pearl River Delta, and make better use of the spatial spillover effect of an industrial layout among cities to realize innovation. The positive role of the government is required to improve the level of urban innovation. It requires that enterprises are encouraged to connect with universities and in-depth exchanges with university scientific research personnel while conducting product technology research and development. The collaborative innovation mechanism guarantees the channels for the transformation of research results and increases the transformation rate of scientific and technological achievements, thereby more efficiently improving the level of urban innovation.

**Author Contributions:** Supervision, H.Z. and E.E.; methodology, Y.Z.; software, Z.S.; validation, H.Z., E.E. and Z.S.; formal analysis, Z.S.; investigation, Z.S.; data curation, Z.S.; writing—original draft preparation, Z.S.; writing—review and editing, Z.S. All authors have read and agreed to the published version of the manuscript.

**Funding:** This research received no external funding.

**Institutional Review Board Statement:** Not applicable.

**Informed Consent Statement:** Not applicable.

**Data Availability Statement:** The data are not publicly available due to restrictions privacy.

**Conflicts of Interest:** The authors declare no conflict of interest.

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
