# Peer review of "Internet Development, Level of Industrial Synergy, and Urban Innovation"

_sustainability, doi:10.3390/su132212410_

Round 1

Reviewer 1 Report

The given report on paper (Reference Number 1451448) is prepared on a request of the editorial board of Sustainability.
Recommendations: I appreciate the authors and would suggest them to address the following points. 

-Rephrase the first line of the second paragraph i.e, The productivity of the industry depends on labor and capital change. 
-Remove the research gape from the subheading 1.3 and write as Industrial collaboration and innovation.
-Correct the subheading 3.1 as Source of data and sampling procedure.
-Once you defined the abbreviation of Gross Domestic Production then you don’t have to repeat the expansion form of GDP in the paper. 
-It is more good to use two digits after decimal in tables. 
-All the results and discussion are fine and covered the main objectives of the paper. 
-The policy implications must be narrow down. 

-At page-5, equations 1-2 are miss aligned in the PDF/generated file, take care of it.

-At start of introduction, second line, authors should highlight different aspects of economic growths. Along with the cited work some other areas of economic growth can also be added. like: economic growth in optical networks that could provide high speed, cheaper, and safer internet technology for high data rate transmission. (a-https://www.sciencedirect.com/science/article/pii/S2211379721000334?via%3Dihub; b-https://ieeexplore.ieee.org/document/9076657; c-https://ieeexplore.ieee.org/document/9576710 etc.

Author Response

Response to Reviewer 1 Comments

First of all, thank you very much for reviewing my article. After reading your modification opinions, I find that there are many problems in my article Your suggestions are of great help to my article. I will modify the article as follows according to your opinions:

Point 1: -Rephrase the first line of the second paragraph i.e, The productivity of the industry depends on labor and capital change. 
-Remove the research gape from the subheading 1.3 and write as Industrial collaboration and innovation.
-Correct the subheading 3.1 as Source of data and sampling procedure.

-At page-5, equations 1-2 are miss aligned in the PDF/generated file, take care of it.

Response 1: These three suggestions are of the same type and are very accurate, so I changed the two subheadings and the first line of the second paragraph as suggested,and adjusted the formula format to align it

Point 2: Once you defined the abbreviation of Gross Domestic Production then you don’t have to repeat the expansion form of GDP in the paper. 

Response 2: This suggestion is also very accurate, so I abbreviate the GDP in Table 1 of the article.

Point 3: It is more good to use two digits after decimal in tables. 

Response 3: According to your suggestion, I change all the data in tables 3, 4 and 5 to two digits after decimal.

Point 4:-The policy implications must be narrow down.

Response 4: According to your suggestion, in the conclusion part, take urban agglomeration as an example to add the specific impact of the policy. The specific contents are as follows: For example, Cities around the Yellow River and other waters, the government should encourage complementary advantages with surrounding cities, learn from the urban development model of the Yangtze River Delta and the Pearl River Delta, and make better use of the spatial spillover effect of industrial layout among cities to realize innovation.

Thank you again for your valuable comments, which will be very helpful to my article.

Reviewer 2 Report

The paper which is assigned to me for review is good but I will recommend correcting some issues. 
Few issues need to be corrected-
-Title i.e., Internet development, level of industrial synergy, and urban innovation. I changed one keyword in the title. 
- The degree of deepening and integration of urban industries will not only improve urban production efficiency but also be an important driving force for urban innovation. What types of innovation, write here some examples. 
- To the best of our knowledge, limited studies have….(This will be more appropriate to write). You cannot just be strongly confident that no study has been conducted on this topic. 
-Adding structure of article at the end of subsection 1.3. is more appropriate. 
-Line 171, level of urban innovation..
-Line 172. Following Pakes and Schankerman [51] the index was used for the patent renewal model as…
-Line 175 replace then with afterward
-Line 177 Most of the internet has been used….
-Line 178 replace “is used” with “has been used”
-Line 194, please assign numbers to all aspects
-Use one spelling for labor either labour or labor in the whole paper
-Line 204 write as “C” and “D”
-Line 208 write C as “C”. A notation should be enclosed with upper double colons.  
-Line 218, “D”. Although I won’t repeat it again, a notation should be enclosed with upper double colons.  
-Write full variables in table 3.
-Write full variables in table 4. 
-Write full variables in table 5. 
-Line 333 the GMM method remained more appropriate and accurate …
-Line 335 and 336, After using GMM, the results of each variable and the main estimation have not been changed (Use the suggested line).
-Line 344 explanatory variables..

Author Response

Response to Reviewer 2 Comments

First of all, thank you very much for reviewing my article. After reading your modification opinions, I found that there are some deficiencies in the article. Your suggestions are of great help to my article. I will modify the article as follows according to your opinions:

Point 1: -Title i.e., Internet development, level of industrial synergy, and urban innovation. I changed  one keyword in the title. 

-Line 171, level of urban innovation..
-Line 172. Following Pakes and Schankerman [51] the index was used for the patent renewal model as…
-Line 175 replace then with afterward
-Line 177 Most of the internet has been used….
-Line 178 replace “is used” with “has been used”

-Line 333 the GMM method remained more appropriate and accurate …
-Line 335 and 336, After using GMM, the results of each variable and the main estimation have not been changed (Use the suggested line).
-Line 344 explanatory variables..

Response 1: These nine suggestions are of the same type and are very accurate, so I corrected the above parts of the article according to your suggestions and change the title to "Internet development, level of industrial synergy and urban innovation", which is more standardized and accurate.

Point 2: -Write full variables in table 3.
-Write full variables in table 4. 
-Write full variables in table 5. 

Response 2: This suggestion is also very accurate, so I have completed the variables in Table 3, 4 and 5

Point 3: - To the best of our knowledge, limited studies have….(This will be more appropriate to write). You cannot just be strongly confident that no study has been conducted on this topic. 

Response 3: Your suggestion is very helpful.The expression in the original text is really inappropriate, so I changed "no study" in the article to "limited"

Point 4:- The degree of deepening and integration of urban industries will not only improve urban production efficiency but also be an important driving force for urban innovation. What types of innovation, write here some examples.

Response 4: According to your suggestion, I added “For example, the sharing mechanism and technology spillover effect formed by the di-vision and cooperation of industries in cities can promote the exchange of knowledge and technology among industries and promote the technological upgrading among industries, so as to realize the innovation of urban scientific and technological level. ” as an example.

Point 5:-Line 204 write as “C” and “D”
               -Line 208 write C as “C”. A notation should be enclosed with upper double colons.  
               -Line 218, “D”. Although I won’t repeat it again, a notation should be enclosed with upper double colons.  

Response 5: This format problem should not appear in the article and has been corrected now.

Thank you again for your valuable comments, which will be very helpful to my article.
